# Whole-brain radiomics for clustered federated personalization in brain tumor segmentation

**Matthis Manthe**[1,2]                        MATTHIS.MANTHE@INSA-LYON.FR
[1] *Univ Lyon, CNRS, INSA Lyon, UCBL, Inserm, CREATIS UMR 5220, U1294, F-69621 Villeurbanne, France*
[2] *Univ Lyon, INSA Lyon, CNRS, UCBL, Centrale Lyon, Univ Lyon 2, LIRIS, UMR5205, F-69621 Villeurbanne, France*

**Stefan Duffner**[2]                             STEFAN.DUFFNER@INSA-LYON.FR
**Carole Lartizien**[1]                     CAROLE.LARTIZIEN@CREATIS.INSA-LYON.FR

**Editors:** Accepted for publication at MIDL 2023

## Abstract

Federated learning and its application to medical image segmentation have recently become a popular research topic. This training paradigm suffers from statistical heterogeneity between participating institutions' local datasets, incurring convergence slowdown as well as potential accuracy loss compared to classical training. To mitigate this effect, federated personalization emerged as the federated optimization of one model per institution. We propose a novel personalization algorithm tailored to the feature shift induced by the usage of different scanners and acquisition parameters by different institutions. This method is the first to account for both inter and intra-institution feature shift (multiple scanners used in a single institution). It is based on the computation, within each centre, of a series of radiomic features capturing the global texture of each 3D image volume, followed by a clustering analysis pooling all feature vectors transferred from the local institutions to the central server. Each computed clustered decentralized dataset (potentially including data from different institutions) then serves to finetune a global model obtained through classical federated learning. We validate our approach on the Federated Brain Tumor Segmentation 2022 Challenge dataset (FeTS2022). Our code is available at (https://github.com/MatthisManthe/radiomics_CFFL).

**Keywords:** Federated learning, Federated personalization, Segmentation, Brain tumor segmentation.

## 1. Introduction

Deep learning methods have shown significant success on a variety of medical image segmentation tasks (Liu et al., 2021b; Futrega et al., 2022). These methods require a large quantity of data to perform well. The construction of inter-institution datasets is constrained by data regulations and overall sensitivity of health data.

Federated learning has been intensively studied within the last years in medical imaging (Li et al., 2019; Liu et al., 2021a; Xu et al., 2022). This paradigm designates training of a machine learning model on a decentralized dataset, enabling the collaboration of different institutions to train a model without sharing data. Convergence speed and final accuracy of federally trained models can however be weakened by the statistical heterogeneity of local datasets. In that sense, multiple ideas have been proposed to improve robustness of the standard *Federated Averaging (FedAvg)* algorithm (McMahan et al., 2017) reducing the

required number of communication rounds and bringing models' accuracy closer to centralized performance (Li et al., 2020; Karimireddy et al., 2020; Wang et al., 2020b; Tang et al., 2022).

Personalized federated learning has been recently introduced as training one model per specific institution while benefiting from others. Main axes of research in this domain revolve around training one model per participating institution through adaptations of meta-learning (Fallah et al., 2020; Acar et al., 2021), multi-task learning (Marfoq et al., 2021), leveraging partial model sharing (Arivazhagan et al., 2019; Pillutla et al., 2022), local finetuning (Li et al., 2021; Yu et al., 2022) or hypernetworks (Shamsian et al., 2021). Clustered federated learning has also been proposed (Ghosh et al., 2020; Sattler et al., 2021) as clustering institutions with similar local distribution and building one model per cluster.

We propose a novel personalization technique tailored to medical image segmentation. In realistic applications of federated learning, participating institutions use different acquisition methods (scanners, acquisition parameters, ...) inducing a feature shift between local datasets. We focus on developing a method specifically for this type of heterogeneity. Furthermore, state-of-the-art methods hypothesize homogeneous local distribution associated to an institution, which is not necessarily the case as an institution can use multiple scanners or vary the acquisition methods depending on the situation. We introduce the idea of sample-level clustered federated learning accounting for both inter and intra-institution heterogeneity while limiting the amount of transmitted information, preserving as much as possible data privacy. Our method enables to build a model for each isolated type of image volume appearance. We apply and validate our approach on the task of brain tumor segmentation based on the FeTS challenge dataset (based on BraTS challenge dataset) in which both inter and intra-institution feature shifts could be verified.

## 2. Method

An overview of the proposed method is depicted on Figure 1. Each institution computes a set of features (first and second order intensity statistics) on each multimodal volume, and sends these feature vectors to the server. The latter normalizes each feature of each received vector and computes clusters in this normalized radiomic feature space. In parallel, classical FedAvg is performed for a certain amount of communication rounds to build an initial global model. Then, FedAvg is performed for each cluster with the previously federally trained model as initialization, giving one final model per cluster of samples with homogeneous texture.

Formally, let $K$ be the number of institutions, each with a local dataset $D_k :=$ $\{(x_{k,i}, y_{k,i})\}_{i=1}^{n_k}$ with $x_{k,i} \in \mathcal{X} = \mathbb{R}^{m \times h \times w \times d}$ the multimodal MRI scan to segment, $y_{k,i} \in$ $\mathcal{Y} = \{0,1\}^{l \times h \times w \times d}$ its associated multi-label ground-truth segmentation map, $n_k$ the local dataset size of institution $k$ and $N = \sum_{k=1}^{K} n_k$ the total number of samples. We note $w \in \mathcal{W} = \mathbb{R}^p$ the parameters of the neural network to be optimized for the downstream task.

**Radiomic features extraction** Each institution extracts a set of radiomic features (first-order and texture features) from each modality and volume. Features of different modalities for a same patient are concatenated into a single feature vector. As opposed to classical approaches with radiomics which try to characterize the texture of a tumor (Shur et al.,

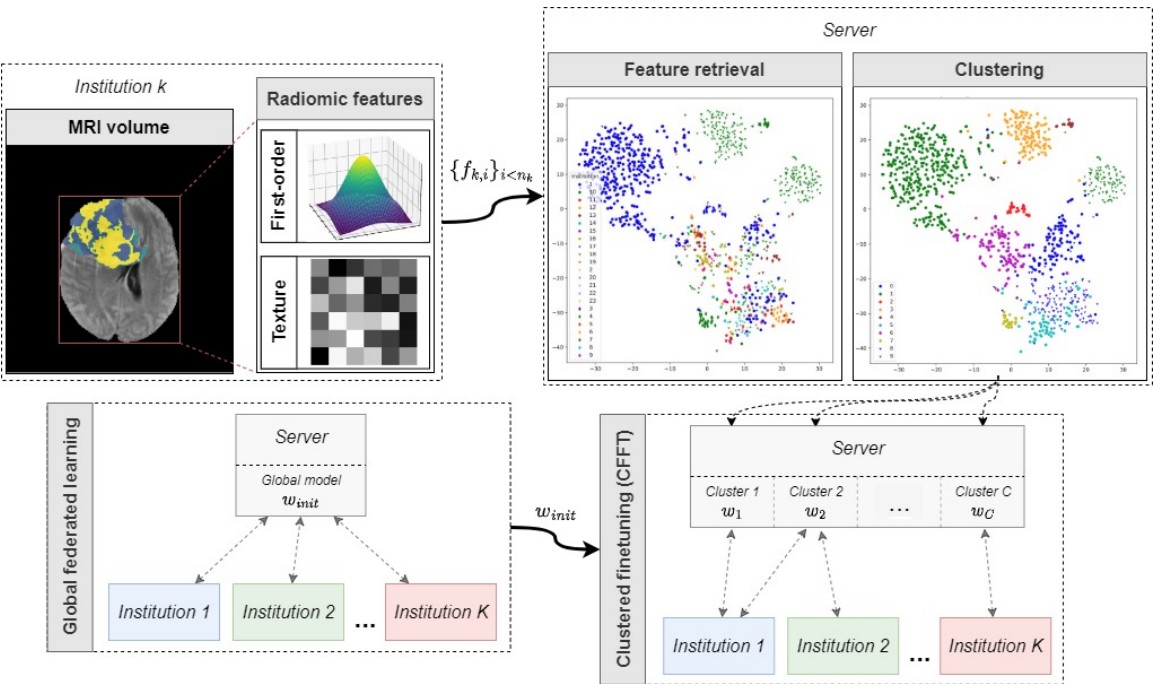

Figure 1: Overall framework of radiomic feature-based clustered federated finetuning.

2021), we compute radiomic features on whole-brain masks on each modality, thus mainly on brain regions which should have a similar appearance with similar acquisition protocols (scanner, parameters, etc.). Formally, for each sample $x_{k,i}$ of institution $k$ a feature vector $f_{k,i} \in \mathbb{R}^R$ is computed and transmitted to the server.

**Server-side clustering**   After retrieving the feature vector associated to each volume of each institution, the server normalizes each feature of each vector. For a given feature with index $r$, we note $f^r$ its value in the feature vector $f$. In this study, the chosen normalization follows Equation (1) with $P^r_{min}$ and $P^r_{max}$ being two chosen percentiles on the pooled set of features across all institutions.

$$\hat{f}^r_{k,i} = max(0, min(1, \frac{f^r_{k,i} - P^r_{min}}{P^r_{max} - P^r_{min}})) \tag{1}$$

To account for highly correlated features, a principal component analysis (PCA) is applied on the set of normalized feature vectors, followed by the computation of $C$ clusters by fitting a Gaussian mixture model (GMM). For training and inference purpose, the normalization parameters, PCA model and GMM are returned to each institution.

We note $Cluster : \mathbb{R}^R \to \{1, ..., C\}$ the application of this clustering process on features extracted from a volume and $\hat{D}_c := \bigcup_{k=1}^{K} \{(x_{k,i}, y_{k,i}) \mid Cluster(\hat{f}_{k,i}) = c, \ i \le n_k\}$ the set of samples assigned to cluster $c$ decentralized over the $K$ institutions. We note $n_{c,k}$ the number of samples of institution $k$ assigned to cluster $c$, and $N_c = \sum_{k=1}^{K} n_{c,k}$ the total size of $\hat{D}_c$.

**Global federated learning initialization** Following (McMahan et al., 2017), we use FedAvg to compute an initial global model $w_{init}$. Given a loss function $l : \mathcal{W} \times \mathcal{X} \times \mathcal{Y} \to \mathbb{R}$, the classical federated objective can be defined as:

$$w^* = \underset{w \in \mathcal{W}}{argmin} \sum_{k=1}^{K} \sum_{i=1}^{n_k} l(w, x_{k,i}, y_{k,i}) . \tag{2}$$

At each communication round $t$, each institution performs $E$ local epoch(s) of stochastic gradient descent (SGD) starting from the current global model $w^t$, giving $K$ updates $\Delta w_k^t = w_k^t - w^t$ to aggregate. The server-side aggregation step is a weighted averaging of these updates $w^{t+1} = w^t + \sum_{k=1}^{K} \frac{n_k}{N} \Delta w_k^t$.

**Clustered federated finetuning** Given the clustering model computed at server-side, we are now able to define a novel clustered federated learning objective:

$$\{w_c^*\}_{c=1}^{C} = \underset{\{w_c\}_{c=1}^{C} \in \mathcal{W}^C}{argmin} \sum_{k=1}^{K} \sum_{i=1}^{n_k} l(w_{Cluster(\hat{f}_{k,i})}, x_{k,i}, y_{k,i}) , \tag{3}$$

where $w_c$ is the parameter set of the global federated model finetuned on dataset $\hat{D}_c$ of cluster $c$. Note that this objective is different from the one defined in recent clustered federated learning approaches such as (Ghosh et al., 2020) as we compute clusters at sample level, not at institution level, enabling to account for an intra-institution heterogeneity. We optimize each cluster model $w_c$ by finetuning the global model $w_{init}$ with FedAvg on the decentralized dataset $\hat{D}_c$ for $T_c$ rounds. The aggregation step for the federated finetuning of cluster model $w_c$ at communication round $t$ becomes $w_c^{t+1} = w_c^t + \sum_{k=1}^{K} \frac{n_{c,k}}{N_c} \Delta w_{c,k}^t$.

**Inference** Given a new sample $x' \in \mathcal{X}$ to segment, the whole clustering pipeline is applied to determine which of the $C$ cluster models must be used. That is, radiomic features $f'$ are extracted from $x'$, normalized following Equation (1), reduced through the computed PCA model, assigned to cluster $c$ using the GMM and segmented with model $w_c$.

## 3. Experiments

### 3.1. Datasets

Experiments were led using *The MICCAI's Federated Brain Tumor Segmentation 2022 Challenge dataset (FeTS2022)* (Bakas et al., 2017; Pati et al., 2021; Reina et al., 2022). This dataset is based on the Brain Tumor Segmentation 2021 Challenge dataset (BraTS2021). Consisting of 1251 multi-modal brain MRI scans (T1, T1ce, T2 and FLAIR) of size $240 \times 240 \times 155$ with isotropic 1mm$^3$ voxel size along with their multi-label tumor segmentation masks including 3 labels, namely enhancing tumor (ET), tumor core (TC) and whole tumor (WT). The real-world partitioning along the 23 acquiring institutions is provided in addition to the samples, enabling to simulate federated learning. Institutions' datasets sizes are very heterogeneous (cf Appendix A.1) with $\sim61\%$ of institutions owning less than 15 samples each. As we do not have the scanner information for every sample, an intra-institution feature shift may exist. This dataset served to evaluate the performance of the proposed clustered federated personalization method and compare it to state-of-the-art methods.

**Dataset splitting** In all of the experiments, a ∼70-15-15% train - validation - test split was followed, giving 833 training, 218 validation and 200 test samples. We computed such split institution-wise, giving a local training, validation and test dataset per institution. We refer to the global train, validation and test sets as the aggregation of each local set, preserving the representation of each institution in each partition. Due to the computational cost of training models on FeTS2022 and the limited amount of samples ($< 5$) provided by multiple institutions, we did not perform cross-validation.

**Image volume pre-processing** Each volume was resized to the bounding box containing all brain voxels, padded to a minimum size of 128 on each dimension if necessary. Each modality of each volume's intensities was then standardized to a zero-mean one-variance gaussian distribution to eliminate the feature shift due to absolute intensity values of scanners.

### 3.2. Radiomic feature-based clustering

**Feature extraction and processing** Ninety-three features including first-order statistics as well as higher order statistical features capturing textural information were extracted per modality using Pyradiomics (van Griethuysen et al., 2017), thus leading to a feature vector of dimension 372. The estimation of these textural features derives from the computation of matrices describing the spatial and intensity relationship between each individual pixel and its neighbors in the image, including GLCM, GLRLM, GLSZM, NGTDM and GLDM (Shur et al., 2021). These matrices require to discretize intensity values of the original images. In this study, we chose an absolute discretization technique based on fixing the histogram bin size to 0.09 for feature extraction. Features were extracted on whole brain masks. A list of the extracted features is provided in Appendix B.1. We first normalized the feature vectors following Equation (1) and setting $P_{min}^r$ as the 2nd-centile and $P_{max}^r$ as the 98th-centile. Dimension of the normalized feature vectors were then reduced to 30 through PCA, preserving 96.58% of the variance. We validated the radiomic features extraction on *The Calgary-Campinas-359 (CC359) dataset* (cf Appendix C).

**Clustering** We fitted a GMM by setting the number of clusters to 10 on the FeTS2022 samples with tied covariance matrix to account for the limited amount of samples compared to the number of remaining feature dimensions. The performance of the proposed radiomic-based clustering method is assessed visually based on the comparison of 2 t-SNE plots of the radiomic feature distribution where each sample is colored either by the label of its belonging institution or by the label of the cluster it was assigned to through the GMM.

### 3.3. Clustered federated finetuning (CFFT)

We evaluated the performance of the proposed federated personalization method for the multi-class segmentation task of the FeTS2022 brain MRI dataset based on standard DICE score and 95% Hausdorff distances.

**Model architecture and data preprocessing** The backbone architecture used in all experiments is a small-sized 3D U-Net (cf Appendix D) trained on 3D patches of size $128 \times 128 \times 128$ with a batch size of 1. We used instance normalization without any learned

parameters to make the model as robust as possible to any feature shift. If not specified, training includes data augmentation focused on reducing the feature shift (gaussian noise, gaussian smoothing, intensity scaling and gamma contrast adjustment).

**Comparison with baseline methods**   We validated our approach against different baselines. We first trained a global model on the pooled FeTS2022 dataset (e.g. BraTS2021) with SGD, referred to *Centralized* in the following. *FedAvg* was used as the baseline global federated optimization algorithm, as described in Section 2. As a personalized FL baseline, we finetuned the FedAvg final model on each local dataset with SGD; this method is referred to as *Local Finetuning*. We then experimented with two versions of our *CFFT* method : the proposed *CFFT* version preserving the privacy of each clustered dataset $\hat{D}_c$ by finetuning with FedAvg and an *ideal* version referred to as $CFFT_{ideal}$, which consists in finetuning on the pooled datasets $\hat{D}_c$ of each cluster with SGD.

**Training hyperparameters**   In *Centralized*, *Local Finetuning* and pooled finetuning of $CFFT_{ideal}$, a learning rate of 0.02 gave the best results. For federated counterparts, a learning rate of 0.05 was used. A weight decay of $10^{-5}$ was used in all experiments, motivated by state-of-the-art publications (Wang et al., 2020a; Yuan et al., 2022). *Centralized* training was performed for 300 epochs, *FedAvg* for 300 communication rounds with one local epoch per round, *Local Finetuning* for 20 local epochs, $CFFT_{ideal}$ for 50 epochs and *CFFT* for 50 communication rounds with one local epoch. The best models at each epoch/communication round were selected based on the validation set performance for all methods. Best results were found by removing data augmentation for clustered finetuning methods, while keeping it for *Local Finetuning*.

## 4. Results and Discussion

### 4.1. Performance of the radiomic features based clustering

We show on Figure 2(a) a t-SNE plot of the radiomic features computed on sample images of FeTS2022. It highlights both inter and intra-institution feature shift in this dataset. We do not have access to the scanner type (vendor, magnetic field) or acquisition parameter to establish further correlation with the observed clusters, but a visual analysis of some example 3D images belonging to different clusters highlights some pattern discrepancies. For example, within institution 4, we can distinguish two types of multi-modal volume appearance (cf Appendix B.2), confirming the existence of intra-institution feature shift, which is well captured by the proposed radiomic based GMM model (Figure 2(b)). Although out of the scope of this paper, we emphasize that feature normalization enabled to spot some outlier volumes (cf Appendix E).

### 4.2. Performance of the clustered federated finetuning (CFFT) method

We show in Table 1 and Table 2 aggregated test DICE scores and 95% Hausdorff distances respectively of *Centralized*, *FedAvg*, *Local Finetuning*, $CFFT_{ideal}$ and *CFFT*, with samplewise standard deviation. Per institution results are given in Appendix A.2. On average *Centralized* training remains the gold standard with the best performance on both metrics. *FedAvg* produces a less robust model, with a gap of more than one DICE point compared to

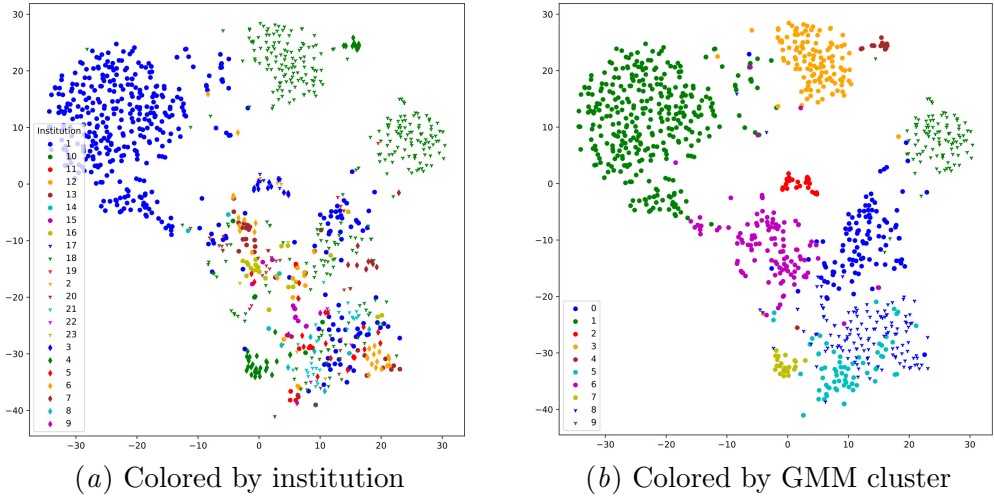

($a$) Colored by institution        ($b$) Colored by GMM cluster

Figure 2: t-SNE plots of radiomic features computed on FeTS2022 samples.

*Centralized* on average. This was shown significant based on a two-tailed Wilcoxon signed-rank test for both metrics with $p < 0.01$. All personalization methods also significantly improve the *FedAvg* DICE score in a similar fashion, with $CFFT_{ideal}$ giving a slight edge on average 95% Hausdorff distance. Note that $CFFT_{ideal}$ performs better than its federated counterpart *CFFT* as we could expect. The goal of this preliminary method is to produce clusters of sample with homogeneous appearance and texture. In that sense, the gap on average metrics between $CFFT_{ideal}$ and *CFFT* is smaller that between *Centralized* and *FedAvg*, motivating a reduction of the shift between institutions taking part in FedAvg in each cluster. Moreover, Figure 3 shows the label distributions of each computed cluster. They are relatively homogeneous, confirming the focus on the feature shift. Other types of shift such as label or concept shifts can be present in the dataset, and we give a hint on their existence in Appendix A.1 with relatively heterogeneous label distributions per institution. Our method's performance is on par with *Local Finetuning* while restricting its effect on feature shift. We emphasize that the proposed paradigm of targeting an identified type of shift would potentially generalize better to unseen data.

On a more general note, the choice of the number of clusters is important. Using too few does not capture feature shift, using too many overfits on clustered training data. Ten clusters gave the best validation results. We must acknowledge the relatively high variability of clustering results, possibly due to the high number of dimensions remaining after PCA to preserve sufficient variance. It must also be noted that the high heterogeneity of the local dataset sizes poses problems to assess the performance of personalized methods, with some institutions only owning one or two test samples (cf Appendix A.2). Finally, we hypothesize in this preliminary work that the computed radiomic features are safe to share with a server. We provide a basic validation of the impracticability of reconstructing a volume based on the 93 features extracted per modality in Appendix F, but it remains an open question as to how these features could be leveraged by an attack against our method. Thus, this preliminary work could be improved by studying the compatibility of the

framework with known privacy-preserving federated learning techniques such as differential privacy or homomorphic encryption, or developing a federated framework including PCA and clustering. Limiting the amount of computed features to the most relevant ones could also be beneficial while opening explainability opportunities.

| Training algorithm | Average | TC | WT | ET |
|---|---|---|---|---|
| Centralized | 0.8912 ± 0.1201 | 0.8896 ± 0.1878 | 0.9188 ± 0.0859 | 0.8651 ± 0.1956 |
| FedAvg | 0.8803 ± 0.1414 | 0.8722 ± 0.2251 | 0.9099 ± 0.0906 | 0.8588 ± 0.2005 |
| Local finetuning | 0.8879 ± 0.1202 | 0.8876 ± 0.1863 | 0.9132 ± 0.0891 | 0.8629 ± 0.1960 |
| CFFT$_{ideal}$ | 0.8887 ± 0.1239 | 0.8867 ± 0.1891 | 0.9139 ± 0.0892 | 0.8654 ± 0.1960 |
| CFFT | 0.8874 ± 0.1254 | 0.8799 ± 0.2056 | 0.9120 ± 0.0907 | 0.8704 ± 0.1852 |

Table 1: Test DICE scores (mean ± standard deviation)

| Training algorithm | Average | TC | WT | ET |
|---|---|---|---|---|
| Centralized | 5.1348 ± 6.2212 | 4.4407 ± 7.2052 | 7.0318 ± 11.0671 | 3.7770 ± 8.5314 |
| FedAvg | 5.8854 ± 7.7368 | 4.7745 ± 8.0444 | 8.8814 ± 15.9499 | 3.7091 ± 7.6837 |
| Local finetuning | 5.9334 ± 7.9817 | 4.8587 ± 9.3901 | 8.9284 ± 16.3270 | 3.5861 ± 7.5650 |
| CFFT$_{ideal}$ | 5.5915 ± 7.2588 | 4.6436 ± 7.3390 | 8.3688 ± 15.3738 | 3.5383 ± 7.5576 |
| CFFT | 5.8749 ± 7.6108 | 4.7359 ± 7.4458 | 8.8709 ± 16.4791 | 3.6520 ± 7.6291 |

Table 2: Test 95% Hausdorff distances (mean ± standard deviation)

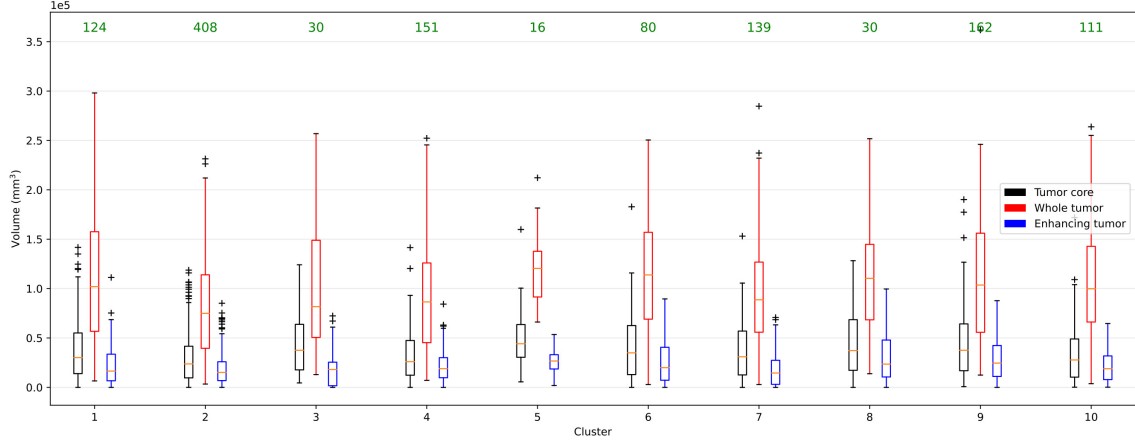

Figure 3: Label distribution per computed cluster in FeTS2022. Green values correspond to the amount of samples associated to each cluster.

## Acknowledgments

This work was partially supported by the Agence Nationale de la Recherche under grant ANR-20-THIA-0007 (IADoc@UdL). It was granted access to the HPC resources of IDRIS under the allocation 2022-AD011013327 made by GENCI.

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

# Appendix A. Per institution analysis

## A.1. Per institution label distribution and local datasets sizes

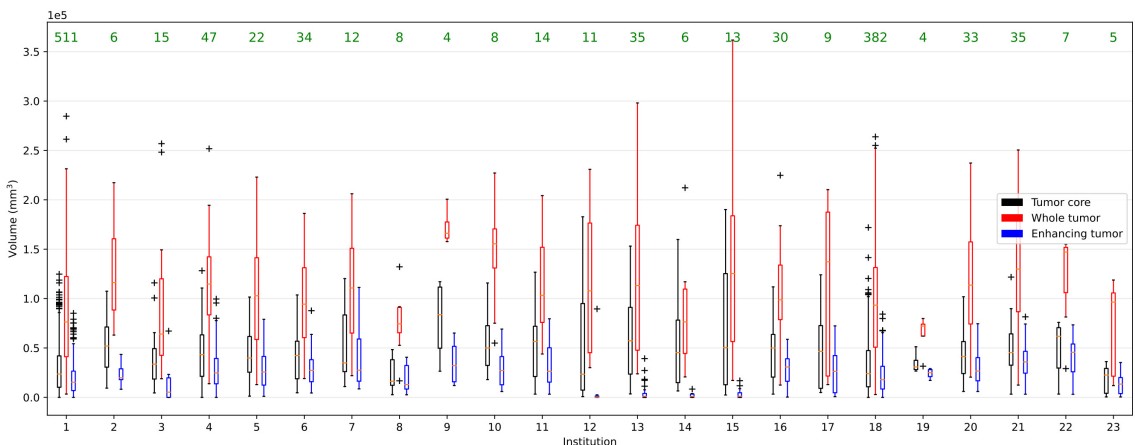

Figure 4: Label distribution per institution in FeTS2022. Green values correspond to the amount of samples associated to each institution in the complete dataset.

## A.2. Per institution results

| Institution | Test size | Centralized | FedAvg | Local Finetuning | $\mathbf{CFFT}_{ideal}$ | CFFT |
|---|---|---|---|---|---|---|
| 1 | 77 | 0.9300 | 0.9256 | 0.9280 | 0.9289 | 0.9285 |
| 10 | 2 | 0.8044 | 0.8051 | 0.8076 | 0.8085 | 0.8072 |
| 11 | 3 | 0.9069 | 0.9000 | 0.9006 | 0.9055 | 0.9053 |
| 12 | 2 | 0.7386 | 0.7031 | 0.7131 | 0.7032 | 0.7106 |
| 13 | 6 | 0.6248 | 0.4910 | 0.6090 | 0.5596 | 0.5781 |
| 14 | 1 | 0.6261 | 0.6115 | 0.6237 | 0.6195 | 0.6100 |
| 15 | 2 | 0.5924 | 0.5282 | 0.6296 | 0.6135 | 0.5539 |
| 16 | 5 | 0.9185 | 0.9211 | 0.9265 | 0.9028 | 0.9095 |
| 17 | 2 | 0.9162 | 0.9133 | 0.9133 | 0.9140 | 0.9101 |
| 18 | 58 | 0.9187 | 0.9117 | 0.9082 | 0.9138 | 0.9119 |
| 19 | 1 | 0.9659 | 0.9625 | 0.9634 | 0.9649 | 0.9650 |
| 2 | 1 | 0.9231 | 0.9221 | 0.9221 | 0.9206 | 0.9160 |
| 20 | 5 | 0.9386 | 0.9314 | 0.9354 | 0.9346 | 0.9358 |
| 21 | 6 | 0.9594 | 0.9545 | 0.9559 | 0.9574 | 0.9599 |
| 22 | 2 | 0.9600 | 0.9553 | 0.9553 | 0.9563 | 0.9570 |
| 23 | 1 | 0.8881 | 0.8919 | 0.8937 | 0.8879 | 0.8867 |
| 3 | 3 | 0.6956 | 0.5952 | 0.6946 | 0.6928 | 0.6681 |
| 4 | 8 | 0.6687 | 0.6669 | 0.7005 | 0.6960 | 0.6943 |
| 5 | 4 | 0.8057 | 0.7946 | 0.7925 | 0.8349 | 0.8204 |
| 6 | 6 | 0.8762 | 0.8762 | 0.8762 | 0.8806 | 0.8844 |
| 7 | 2 | 0.8895 | 0.8873 | 0.8924 | 0.8828 | 0.8804 |
| 8 | 2 | 0.9631 | 0.9621 | 0.9621 | 0.9645 | 0.9634 |
| 9 | 1 | 0.7895 | 0.8232 | 0.7818 | 0.8545 | 0.8416 |

Table 3: Average test dice scores per institution

| Institution | Test size | Centralized | FedAvg | Local Finetuning | $\text{CFFT}_{ideal}$ | CFFT |
|---|---|---|---|---|---|---|
| 1 | 77 | 3.6752 | 4.3375 | 4.5268 | 4.3651 | 4.4183 |
| 10 | 2 | 7.5957 | 7.1775 | 7.1275 | 6.8905 | 7.1512 |
| 11 | 3 | 3.1412 | 3.5267 | 3.2199 | 3.2042 | 3.0589 |
| 12 | 2 | 10.6487 | 24.9708 | 26.7511 | 23.4790 | 28.9451 |
| 13 | 6 | 8.1182 | 12.2476 | 13.0226 | 8.0737 | 10.0930 |
| 14 | 1 | 3.2863 | 4.2002 | 3.3708 | 4.1926 | 5.0000 |
| 15 | 2 | 15.2838 | 17.3938 | 18.8894 | 19.4479 | 24.4377 |
| 16 | 5 | 4.3992 | 8.8826 | 8.7169 | 9.8859 | 9.9735 |
| 17 | 2 | 1.7756 | 2.0977 | 2.0977 | 1.9249 | 2.2928 |
| 18 | 58 | 4.7928 | 5.1874 | 5.0459 | 5.1053 | 4.8157 |
| 19 | 1 | 1.6667 | 1.3333 | 1.3333 | 1.1381 | 1.1381 |
| 2 | 1 | 5.0726 | 5.9127 | 5.9127 | 6.1498 | 8.7888 |
| 20 | 5 | 5.8938 | 2.6021 | 2.4663 | 2.4494 | 2.3830 |
| 21 | 6 | 2.1143 | 2.2983 | 2.3060 | 1.6677 | 1.5816 |
| 22 | 2 | 2.4694 | 3.1271 | 3.1271 | 3.2594 | 15.0383 |
| 23 | 1 | 4.2356 | 3.9560 | 3.7742 | 4.2201 | 4.2560 |
| 3 | 3 | 6.5017 | 9.2293 | 7.6449 | 7.4780 | 8.9391 |
| 4 | 8 | 14.1589 | 12.5674 | 12.6770 | 10.3157 | 10.3217 |
| 5 | 4 | 5.2451 | 7.8477 | 7.7993 | 6.7148 | 7.6399 |
| 6 | 6 | 10.0190 | 10.2264 | 10.2264 | 9.6674 | 9.5711 |
| 7 | 2 | 12.2938 | 15.7393 | 14.9248 | 16.0055 | 16.4357 |
| 8 | 2 | 1.2761 | 1.4267 | 1.4267 | 1.3291 | 1.3738 |
| 9 | 1 | 12.8816 | 6.7070 | 7.9211 | 6.1545 | 6.5459 |

Table 4: Average test hausdorff distances per institution

# Appendix B. Radiomic features extraction

## B.1. Radiomic features list

Using Pyradiomics, 93 features are extracted per modality:

- First Order Statistics (18 features) (standard deviation is not included)

- Gray Level Cooccurence Matrix (GLCM) (24 features)

- Gray Level Run Length Matrix (GLRLM) (16 features)

- Gray Level Size Zone Matrix (GLSZM) (16 features)

- Neighbouring Gray Tone Difference Matrix (NGTDM) (5 features)

- Gray Level Dependence Matrix (GLDM) (14 features)

## B.2. Visual examples

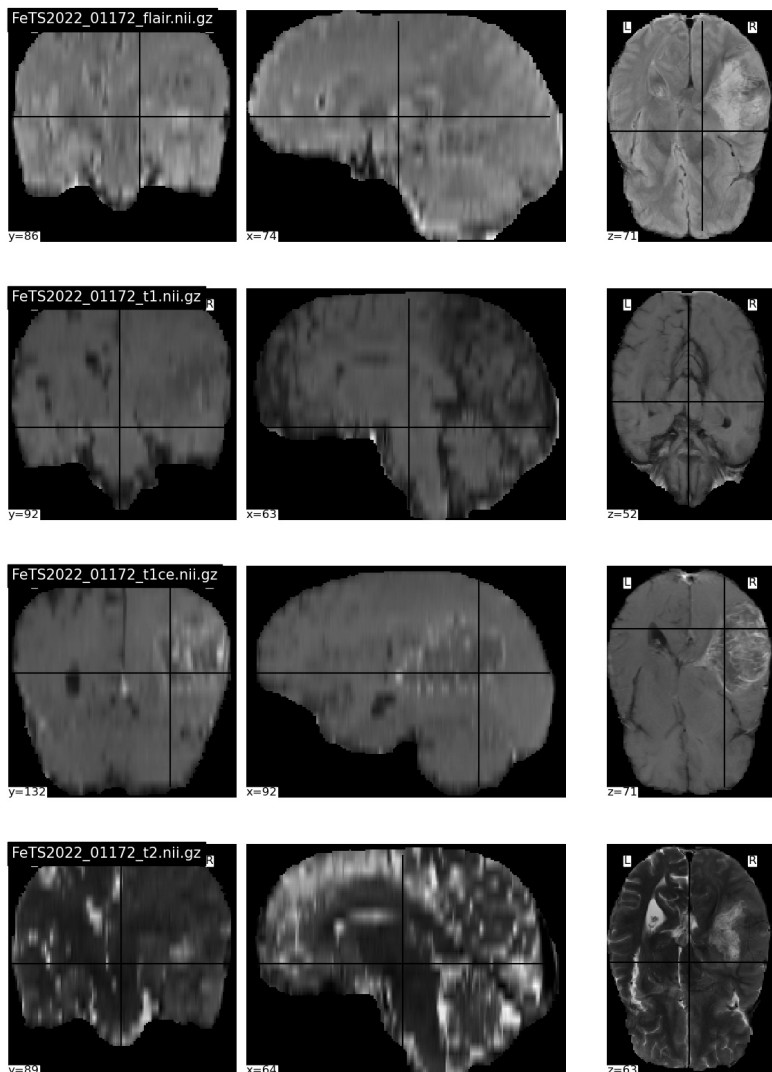

Figure 5: FeTS2022_01172 MRI scans, owned by institution 4, assigned to cluster 7.

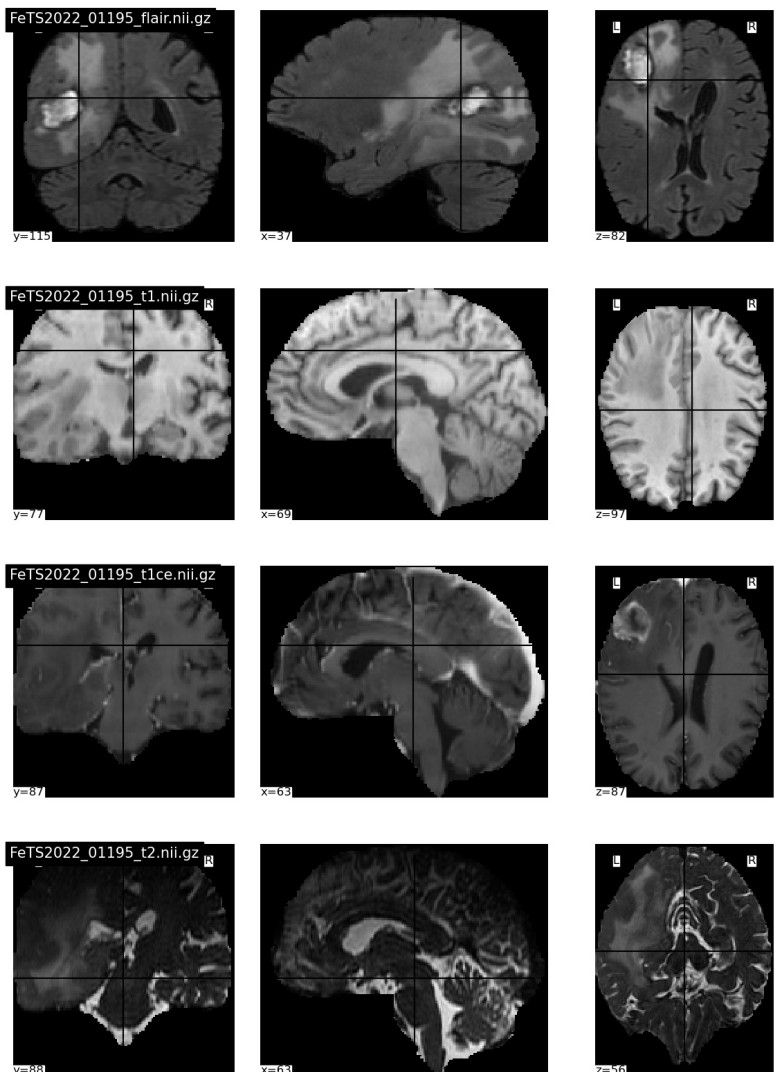

Figure 6: FeTS2022_01195 MRI scans, owned by institution 4, assigned to cluster 4.

## Appendix C. Radiomic features validation on CC359

**Dataset** *The Calgary-Campinas-359 (CC359) dataset* (Souza et al., 2018). This dataset consists of 359 T1-weighted brain MRI scans of healthy subjects along with their "silver-standards" brain masks generated both using the STAPLE algorithm. These images were acquired on scanners from three vendors (Siemens, Philips and General Electric) at both 1.5 T and 3 T. Sixty exams were acquired per vendor and magnetic field strength, except for Philips 1.5T which totalizes 59 exams. This dataset, concatenating 6 series of exams of equal size and low intra-feature shift serves as a use case to validate the radiomic feature extraction process.

**Parameters** We fixed the histogram bin size to 0.15 for features extraction on the CC359 dataset. We normalized the feature vectors following Equation (1) and setting $P_{min}^r$ as the 2nd-centile and $P_{max}^r$ as the 98th-centile. Dimension of the normalized feature vectors is then reduced to 30 through PCA, preserving 99.96% of the variance.

**Results** We show on Figure 7 a t-SNE plot of the radiomic features computed on CC359 after normalization and PCA where each label encodes for one of the 6 scanners on which the image were acquired (e.g. philips_3 corresponds to a 3T Philips scanner). Despite only dealing with healthy patients with a normalization process that significantly reduces feature shift, distinct clusters can be identified for each scanner manufacturer and magnetic field value. This demonstrates the capacity of the selected radiomic feature vector to capture textural patterns induced by the scanner characteristics.

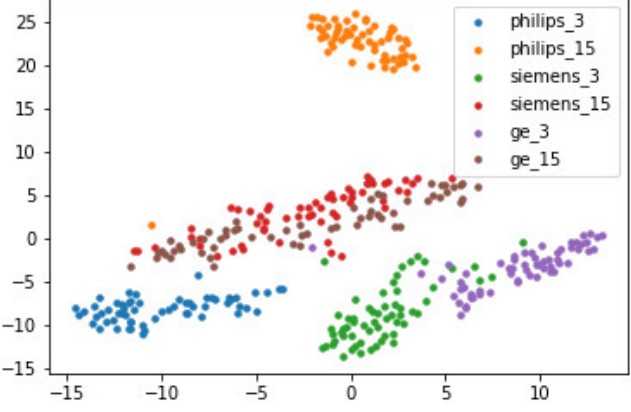

Figure 7: t-SNE of radiomic features.

## Appendix D. Model architecture

We use a small-sized 3D U-Net as the backbone architecture. It includes 3 down-sampling and up-sampling paths. Stride 2 convolutions are used in down-sampling paths, and transpose convolutions for up-sampling paths. Kernel size is (3,3,3) for every convolution block. A first stride 1 convolution block outputs 16 channels, multiplied by 2 at each downsampling path to reach 128 channels in the bottleneck part of the network. Inference are done with a sliding window with an overlap of 0.5 and Gaussian aggregation of results on overlaps.

## Appendix E. Outliers detection

For some extracted features, the normalization process clearly highlighted outliers. As example, the GLCM Cluster Prominence feature computed on the FLAIR n°1441 was two orders of magnitude higher than for any other volume. We verified that this volume has a particular appearance, with extreme intensities toward the eye balls (Figure 8). Such manually computed features can thus be leveraged for federated outlier detection without sacrificing too much privacy.

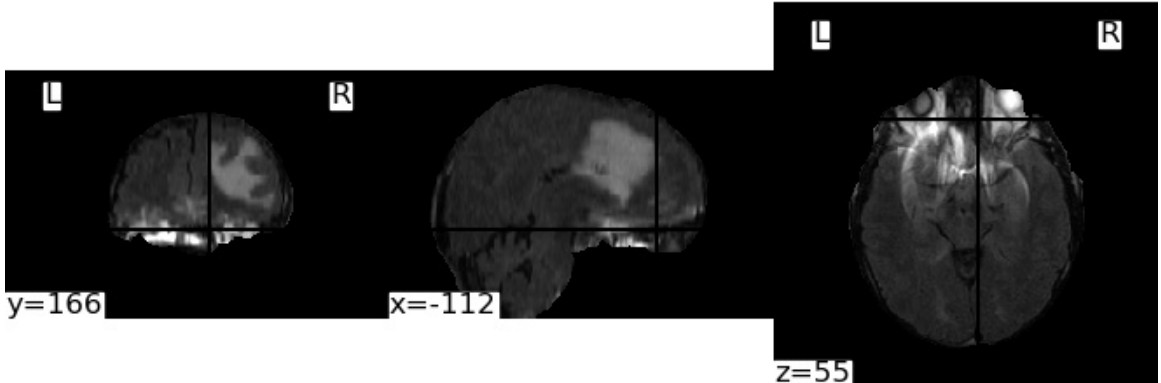

Figure 8: Flair MRI scan of patient n°1441

## Appendix F. Privacy preservation basic validation

We explore in this section the efficiency of an attack on the radiomic features communicated during the proposed clustered federated finetuning process. Its objective is, given every normalized feature vector computed on the train and validation T1 volumes, to train a decoder to reconstruct the associated volumes. Each volume was standardized and resized to $128^3$ voxels to simplify the task.

**Model**  The model is composed of a first linear layer outputting 4096 values reshaped to [512, 2, 2, 2], followed by 6 3D residual convolutional blocks, upsampling each dimension by a factor of 2 while dividing the number of channels by 2. We use 3D batch normalization and LeakyRelu activation function with a slope of 0.2 between each layer. The model is composed of approximately 20 millions parameters.

**Training parameters**  We train this model using the same training - validation - test split as for the original segmentation task. The loss and metric used is a standard MSE. We use Adam with a learning rate of 1e-4 and a weight decay of 1e-5 for 300 epochs with a batch size of 2. The final model is selected based on best validation performance.

**Result**  We show in Figure 9 the training and validation loss curves. After only 20 epochs, the model starts to overfit on the training set with a stagnating validation performance. We show in Figure 10 two slices of reconstruction outputs of the test set. The model is only able to reconstruct an average volume, validating the fact that the transmitted feature vectors do not contain enough information to reconstruct a volume, even the tumorous parts. The proposed scheme of attack is also unrealistically powerful, as it presumes a large amount of already leaked data.

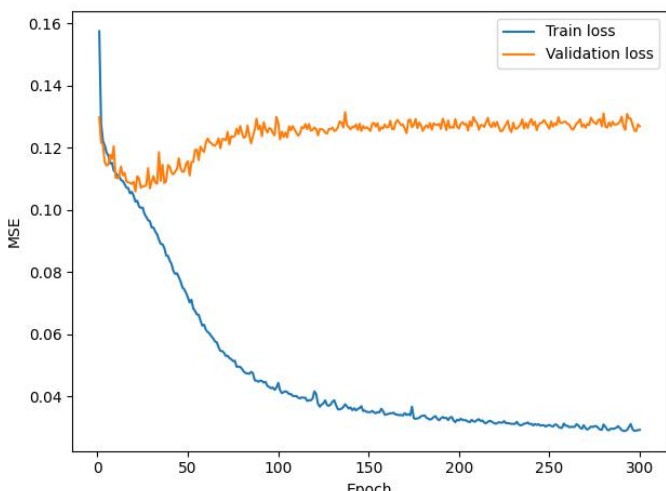

Figure 9: Training and validation loss curves of a model reconstructing T1 volumes based on extracted radiomic features.

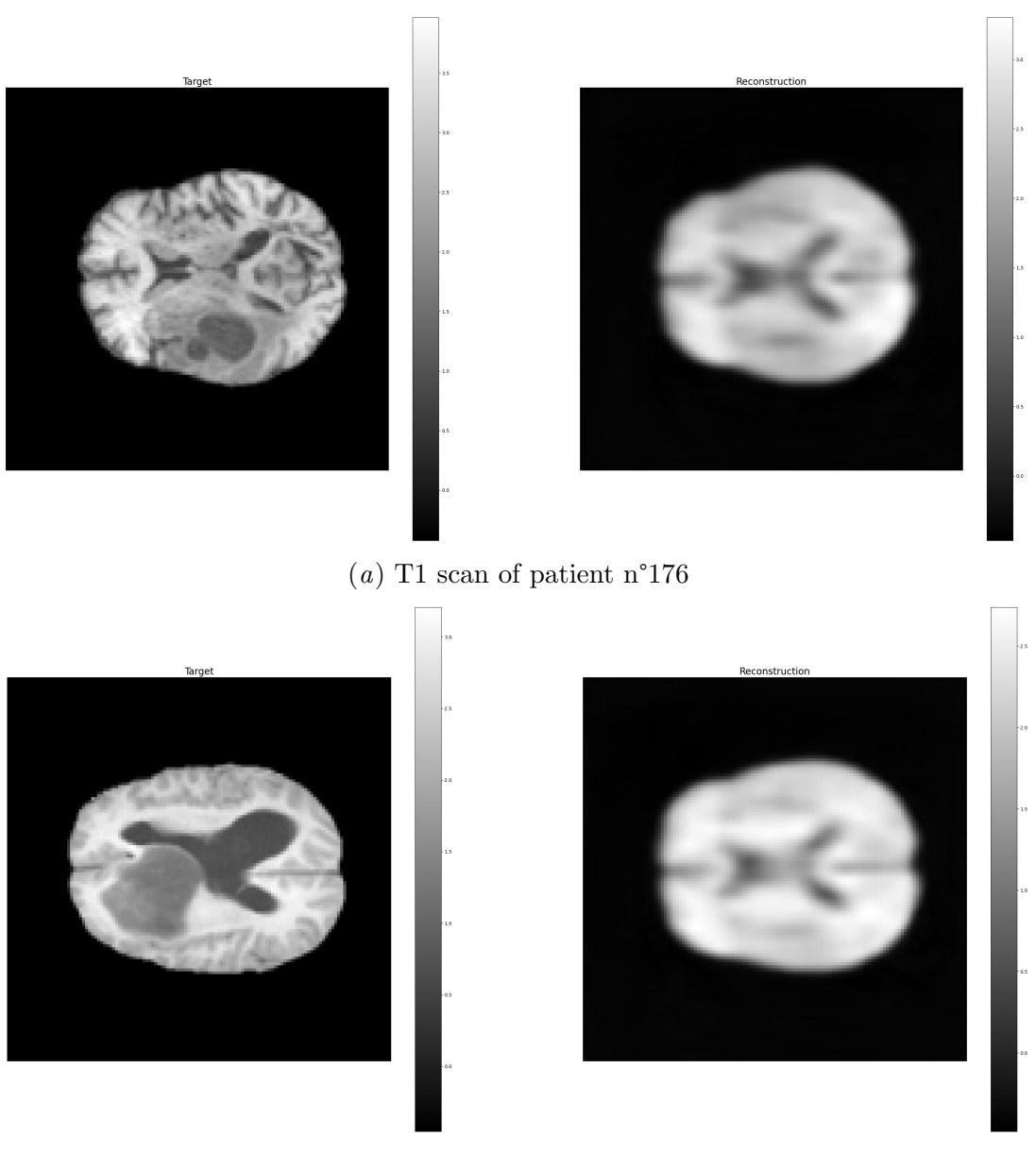

(a) T1 scan of patient n°176

(b) T1 scan of patient n°134

Figure 10: Example outputs of a model trained to reconstruct T1 volumes based on trans-mitted radiomic features. We show slice n°64 for both samples. On the left are ground truth standardized volumes, on the right are reconstructed volumes.

