# OpenReview forum: "Whole brain radiomics for clustered federated personalization in brain tumor segmentation"
_MIDL.io/2023/Conference — MIDL 2023 Poster_

### Official Review · Reviewer_xihx · 2023-01-29

**Confidence:** 3
**Preliminary Rating:** 4
**Recommendation:** Poster

**Summary:**

Federated learning for medical applications can have several applications enabling sharing of data/models between institutions with  privacy-preserving features. This work addresses the feature fluctuations that are due to variations in inter- and intra- institution acquisitions. Instead of having models specific to institutions, images from all clients are clustered at the server side using radiomic features, and models specific to these clusters are used. Experiments on the federated brain tumour segmentation task show that the proposed federation strategy could be beneficial compared to standard federation methods.

**Strengths:**

* The main contribution in this work is to base federation at sample level, instead of using institution/scanners. This has merit as by aggregating all images and normalising them in an identical fashion could capture the most important differences in the data, which can be taken into account when federating.

* The proposed strategy for federation is simple, as it is based on simple feature extraction from the volumes, feature normalisation and then clustering.

* Experiments on the FeTS2022 dataset and the baseline comparisons are reasonable, showing comparable performance.

* The paper is largely clearly written with an easy-to-follow visualisation of the method in Fig. 1.

**Weaknesses:**

**Features used**: One of the key features of this method is to use individual sample level clustering based on simple features extraction. The motivation for using these features is not sufficiently clear. Have the authors tried any other features? Are these sufficiently expressive to capture the variations in the dataset? How was this tested? Perhaps some analysis of the dataset could make for a stronger case.

**Privacy preserving**: In Page 2, the authors say
> limiting the amount of transmitted information, preserving as much as possible data privacy

I am concerned about this loose statement, and more so when it comes to the claims of privacy preservation. What do the authors exactly mean? Later in the text, the authors seem to be more assertive of the privacy preserving claims when presenting the CFFT method compared to the CFFT-ideal method. In conclusions, the authors mention that:

> inferring some properties could be possible

Which specifically?

**Performance improvements**: The performance improvements compared to FedAvg are not large/significant. This in itself is not a problem. It would be useful to discuss this more clearly, as to what other improvements could be done. In my opinion, the feature extraction might be too simple; use of autoencoders with percpetual loss or similar could be more useful.


**Minor comments:**

* FedAvg is an essential component of the proposed method (Eq. 2); I would suggest introducing FedAvg as part of related work and not after presenting the proposed method. Would improve readability.

* w are introduced as parameters of a neural network. Can this be clarified further? Perhaps specify that the neural network for the downstream task, so as to not imply that it is not a neural network used for federation? Again, this is for improving readability.

**Deanonymize Review:**

no

**Detailed Comments:**

See review above.

**Paper Type:**

methodological development

**Questions To Address In The Rebuttal:**

See the weakness above.

1. More discussion on the motivation for the specific set of features used
2. Clarification on the privacy preserving claims
3. Fix the minor issues mentioned for improving readability

---

### Official Review · Reviewer_uUoV · 2023-02-02

**Confidence:** 4
**Preliminary Rating:** 4

**Summary:**

The paper describes a novel approach to segment multimodal MRI scans of brain tumors in a federated learning setting: First, high-dimensional feature vectors are computed for each scan. These feature vectors are pooled globally, across institutions, and then, using a pre-determined number of clusters C, a pre-trained model (obtained via the privacy-preserving well-known FedAvg algorithm) is refined by continued training on each cluster individually. The authors show that this approach gives results that are comparable, and often even slightly superior, to standard federated learning approaches, such as FedAvg. Because feature vectors do not allow for full reconstruction of the scans, the method preserves privacy, albeit not to the extent that FedAvg does as clusters obtained by the method contain data from different institutions. Thus, the proposed method is attractive for studies involving a large number of institutions where, in addition to accuracy, data privacy is a paramount concern and a feature shift is suspected to exist.

**Strengths:**

The clustered federated finetuning (CFFT) algorithm is an interesting and novel approach that is able to detect and partly compensate for inter- and intra-institutional feature shifts. Thus, institutions might benefit from using the personalized cluster-based model parameters, in principle even for each of their scanners. This way, the feature shift between institutions and scanners might be taken into account when processing new data. The method outperforms standard federated learning approaches (FedAvg) on a publicly available dataset.

**Weaknesses:**

The paper is partially hard to understand, for the following reasons:

- Why is it expected that feature vectors from the same institution are close to each other in a t-SNE plot (the main assumption described in the introduction)? Different patients, different scanners, and acquisition protocols would also give rise to heterogeneity, and thus to different clusters even within an institution. The rationale behind this should be explained much better. This is also related to Figs. 3 and 5 in the Appendix, where it is shown that the label distribution per cluster is more homogeneous than the label distribution per institution, which is interpreted as a major advantage of the CFFT method.
- Section ‘Clustering’ on p.5: It should be explained why a simple visual inspection of two t-SNE plots suffices to conclude that the approach is valid. Presumably, because the clusters largely preserve institutional membership, but also clustering together data from different institutions so as to balance the label distributions? This should be made much clearer.
- The authors mention that the number of clusters largely influences the final performance results. Indeed, the fewer clusters there are, the closer the performance should be to the centralized algorithm. Is it really like that? Does the amount of data points the system has seen in each of the clusters in general increase the performance? A guideline on how to choose the best number of clusters and results for different cluster numbers would have been very helpful.
- Is the code available? Please, make a general statement on code and data availability to allow the reproduction of the results.

**Deanonymize Review:**

no

**Detailed Comments:**

- Figure 1 should be improved and described more. What is CLFT? This abbreviation is not used anymore in the paper.
- Figure 3 - green numbers on top - what are those? Presumably the number of scans in each institution?
- p. 2: What are m and l? (Guess: number of modalities, number of labels). Please make sure all symbols are defined.
- p. 3: typo: ‘intial’.
- p. 4: “were led using The MICCAI’s Federated Brain Tumor Segmentation 2022 Challenge dataset (FeTS2022)”. Please, name the dataset consistently throughout the paper.
- p. 6, section ‘Comparison with baseline methods’: There should be different symbols for the two clustered or pooled datasets. Currently, one symbol is used.
- p. 6: CFFT_ideal should be explained in more detail.
- Abstract: ‘To mitigate this effect, federated personalization emerged as the federated optimization of one model per distribution.’: ‘Distribution’ should be changed to ‘institution’.
- Supplementary information should be re-ordered in the order of its first appearance in the text.
- In all figures, please expand the captions, so reading the caption would suffice to understand what different letters and numbers mean.

**Paper Type:**

methodological development

**Questions To Address In The Rebuttal:**

See the comments described in “Weaknesses”; especially point 2: please present more details (and ideally corresponding data) on how the choice of the number of clusters C affects the segmentation accuracy.

---

### Official Review · Reviewer_28u3 · 2023-02-06

**Confidence:** 4
**Preliminary Rating:** 4
**Recommendation:** Poster

**Summary:**

The paper proposes clustering image samples coming from various institutions by using global texture features extracted from the images and then learning image segmentation models for each cluster which are used to finetune a global model in a federated learning setup. The experiments are carried out for brain tumor segmentation using Federated Brain Tumor Segmentation 2022 Challenge dataset.

I've read the comments and rebuttal. It would be better to rephrase to tone down the "seems impossible" point regarding reconstruction from the features used in this method. Although the rebuttal does not address the sample vs institutional clustering comparison, the other elements of the paper are useful and hence I've updated the rating.

**Strengths:**

The paper is well-written and easy to follow.

The idea of learning clusters to group together image samples from across the institutions and learning cluster-wise models is interesting that could open new ways of doing optimal federated learning.

**Weaknesses:**

The features are sent to the server which could lead to violating privacy if the images could be reconstructed from the features. The authors suggest reconstructing images from the features “seems impossible” with current approaches. This is not verified, although some straightforward experiments could be done, such as: a convolutional decoder could be trained to reconstruct images in the given dataset using the features calculated in the proposed method as input and see if the decoder can reconstruct images or not.

The authors suggest that compared to Ghosh et al. 2020, the proposed method has the advantage of computing sample-level clusters to account for intra-institution variability. However, this benefit is not compared in experiments to see sample-level vs. institution-level clustering approaches.

The authors suggest that the features capture scanner and acquisition parameter variability. However, what is the impact of other variabilities such as age (children’s brain vs older adult’s?). It would be better to discuss how the proposed method could handle these other factors of variations.

**Deanonymize Review:**

no

**Detailed Comments:**

The train and test set contains all the institution’s dataset. It would be interesting to see how well it can generalize to unseen new institution’s dataset by doing an experiment where a few institutions’ dataset is kept as an independent test set without using them in a training set.

**Paper Type:**

methodological development

**Questions To Address In The Rebuttal:**

Comparing institutional vs sample level clustering results, discussion on factors of variations other than acquisition parameters, and basic validation of the statement that image reconstruction from the proposed features is “impossible”.

---

### Meta-Review · Area_Chair_yo71 · 2023-02-23

**Recommendation:** Accept (Poster)
**Confidence:** 5

**Metareview:**

This paper proposes a federated learning algorithm to deal with inter and intra-institution heterogeneity in MR scanners, based on clustering of radiomic features. The method is validated using the Federated Brain Tumor Segmentation approach. All reviewers have recommended weak acceptance for this work. I agree with the reviewers which indicate that the paper is well-written and easy to follow, and the clustered federated finetuning algorithm is novel. Thus, after reading the reviewers comments and the rebuttal provided by the authors, I recommend acceptance of this work.

For the camera ready, the authors must take into account the comment raised by reviewer 28u3 which states that “It would be better to rephrase to tone down the "seems impossible" point regarding reconstruction from the features used in this method.”